# Spanish Nurses’ Knowledge and Perceptions of Climate Change: A Qualitative Study

**DOI:** 10.3390/nursrep15070226

**Published:** 2025-06-24

**Authors:** Antonio Miguel Caraballo-Betancort, Irene Marcilla-Toribio, Blanca Notario-Pacheco, Maria Leopolda Moratalla-Cebrian, Ana Perez-Moreno, Alba del Hoyo-Herraiz, Raquel Poyatos-Leon, Maria Martinez-Andres

**Affiliations:** 1Health and Social Research Centre, Universidad de Castilla-La Mancha-Campus de Cuenca, 16071 Cuenca, Spain; antoniomiguel.caraballo@alu.uclm.es (A.M.C.-B.); ana.perez22@alu.uclm.es (A.P.-M.); raquel.poyatos@uclm.es (R.P.-L.); maria.martinezandres@uclm.es (M.M.-A.); 2Research Group Health, Gender, and Social Determinants, Universidad de Castilla-La Mancha-Campus de Cuenca, 16071 Cuenca, Spain; leopolda.moratalla@uclm.es (M.L.M.-C.); albacaterinadel.hoyo@alu.uclm.es (A.d.H.-H.); 3Faculty of Nursing of Cuenca, Universidad de Castilla-La Mancha-Campus de Cuenca, 16071 Cuenca, Spain; 4Faculty of Nursing of Albacete, Universidad de Castilla-La Mancha-Campus de Albacete, 02071 Albacete, Spain

**Keywords:** climate change, climate crisis, nursing education, health literacy, qualitative research

## Abstract

**Background/Objective:** Nurses play a critical role in addressing climate change. They are instrumental in both mitigation and adaptation to its effects. Through care provision, education, management, policy development, and research, nurses can undertake a variety of specific actions in response to climate change. However, their perceptions of this challenge remain under-researched. This study aims to investigate Spanish nurses’ knowledge of climate change and its impact on health. **Methods:** This is a qualitative descriptive study based on the constructivist paradigm. Purposive and snowball sampling strategies were used to recruit nurses from emergency services, geriatrics, cardiology, respiratory medicine and primary care in nine different regions of Spain. Semi-structured online interviews were conducted. Data analysis was carried out by three researchers via a three-stage inductive thematic analysis approach. **Results:** The sample consisted of 31 nurses, predominantly women (77.42%), with a mean age of 41 years. Seventy percent of the participants had less than 15 years of experience in the service. Four categories were identified: (i) general knowledge of climate change; (ii) knowledge of climate change and health; (iii) knowledge of actions to address climate change; and (iv) knowledge development. Overall, the nurses demonstrated awareness of the risks posed by climate change as well as actions to respond. However, barriers such as a lack of formal training and eco-anxiety affect their knowledge acquisition. **Conclusions:** Nurses play an important role in the response to climate change. However, more comprehensive and higher-quality educational programmes, provided by academic institutions, workplaces, and professional associations, are needed. This study was prospectively registered with the Clinical Research Ethics Committee of the Cuenca Health Area on 25 January 2022 (registration number 2021/PI3721).

## 1. Introduction

It is unequivocal that human activity has caused changes in the Earth’s climate [1,2]. These climatic changes have both direct and indirect impacts on human health, affecting millions of people worldwide [2,3,4]. According to van Daalen et al. in The 2024 Europe Report of the Lancet Countdown on Health and Climate Change, “climate change is not a far-in-the future theoretical scenario: it is here, and it kills” [5]. Southern European countries are experiencing more intense climate change effects than other regions of the continent [5]. For instance, this region experienced the highest relative increase in heat vulnerability in Europe—an 11% rise between 1990 and 2022 [5]. In Spain, some consequences of climate change—such as heatwaves [6], floods [7,8], air pollution [9], and the spread of climate-sensitive infectious diseases and their vectors [10,11]—already cause thousands of deaths annually. Owing to its severe health impacts, climate change is regarded as “the biggest global health threat of the 21st century” [12]. In recognition of its public health implications, climate change was formally declared a public health emergency by the Regional Office for Europe of the World Health Organization in July 2023 [4].

According to Watts et al., “tackling climate change could be the greatest global health opportunity of the 21st century” [13]. Nurses, as the largest professional group within the healthcare workforce [14], play a pivotal role in addressing climate change [15]. They are well positioned to lead the response through their work as clinicians, educators, researchers, policy advocates, and healthcare executives [16]. Nurses play a vital and multifaceted role in responding to climate change, both individually and collectively through professional associations. From a planetary health and climate justice perspective, their responsibilities include addressing the health consequences of climate change and strengthening the resilience of health systems [16,17]. Resilient health systems are those capable of absorbing shocks, maintaining core functions, and reorganising effectively in the face of crises such as climate-related events [18]. Nurses are also key actors in education, equipping healthcare professionals and communities with the knowledge required for climate mitigation and adaptation [19]. Moreover, nurses are called to lead sustainability initiatives within healthcare systems and to engage in policy advocacy and public activism [16,19]. They contribute to research and evidence dissemination, and promote sustainable practices and intersectoral collaboration to enhance climate action [16]. Adopting a planetary health framework enables nurses to integrate ecological awareness into professional practice and to advocate for equitable and effective responses to climate threats [17].

To date, several studies—mainly quantitative—have explored nurses’ knowledge of climate change in different countries. The findings of these studies indicate significant variations in the levels of knowledge demonstrated. However, they consistently reflect a strong interest in the subject among nurses [20,21]. Qualitative methodology—particularly within the constructivist paradigm—has demonstrated its value in exploring perceptions from the perspective of the individuals under study [22]. Each person’s perception of climate change is understood as a construct shaped by their contextual circumstances and individual background [23]. In southern Europe, apart from studies involving healthcare professionals without distinguishing by profession [24,25], qualitative methodology has only been applied to explore graduate nurses’ perceptions in Turkey [26]. In Spain, a study conducted by Luque-Alcaraz et al. in 2024 [27] employed a mixed-method approach to examine the environmental awareness of Spanish nurses. The study identified nurses’ perceived lack of competencies related to environmental awareness, particularly regarding knowledge, skills, and attitudes. Despite its relevance to climate change, this study focused specifically on environmental awareness and the efficacy of Green Teams. These Green Teams are multidisciplinary teams, often led by nurses, which promote sustainable practices and carbon footprint reduction within healthcare settings [27].

Therefore, our study is the first to explore the knowledge of graduate nurses in Spain concerning climate change and its health effects, drawing on the depth afforded by qualitative methodology. Thus, the aim of this study was to investigate Spanish nurses’ knowledge of climate change and its impact on health. To achieve this objective, the following research question was posed: What do Spanish nurses know about and how do they perceive climate change and its effects on health?

## 2. Materials and Methods

### 2.1. Design, Participants and Setting

This was a qualitative descriptive study [28,29] based on the paradigm of constructivism [30]. Consistent with this paradigm, the influence of the research team and the contextual setting on the final results was acknowledged. This influence was reflexively managed throughout the entire research process in a continuous, collaborative, and multifaceted manner [31]. The detailed protocol of this study is available for consultation [32].

The sample included nurses from various regions and public healthcare services across Spain. The selected services were primary care, emergency services, geriatrics, cardiology, and respiratory medicine. The selection of these services was made on the basis that they constitute points of entry into the Spanish healthcare system, or due to their association with population groups and health conditions that are most vulnerable to the impacts of climate change [32]. The selected regions were chosen based on their levels of air pollution. The zoning criteria for air quality followed those employed by the Ministry for Ecological Transition and the Demographic Challenge of the Spanish Government, as well as by the European Environment Agency. Air pollution levels were determined according to the concentration of NO_2_ in the atmosphere. NO_2_ concentration records from 2021 were used, as they were the most recent available ones at the time of drafting the research protocol. The regions were categorised into three levels of air pollution—high, medium, or low. Three regions were selected for each pollution level: the three with the highest NO_2_ concentrations, the three with the lowest concentrations, and three with intermediate values [32]. All the participants were adults, had at least one year of experience in the service where they were working at the time of the interview, and were able to express themselves in Spanish and connect to the online interview. Purposive and snowball sampling strategies were used. The assistance of collaborators close to the units of interest was essential in the recruitment of volunteers interested in the subject of this study. Each volunteer was subsequently contacted by email, phone, or text message. The objective of this contact was to explain this study in more detail, confirm the suitability of the volunteer, and schedule a future interview. The sample consisted of 31 nurses who agreed to be interviewed. In addition, contact was lost with eight individuals who had initially expressed an interest in participating in this study. A further four individuals indicated their decision not to proceed due to scheduling constraints. One additional individual was excluded on the basis of being employed in a privately managed healthcare facility.

The interviews were conducted via the Microsoft Teams platform. This allowed the interviewees the flexibility to choose the environment from which to participate. All the interviews were conducted by the same researcher (A.M.C-B). The interviewer was a male nurse with a master’s degree in health and social research and over six years of professional experience. He also had prior experience in qualitative research and in the field of climate change and its impacts on health. His knowledge and professional background facilitated his understanding of the ideas expressed by the interviewees. No personal information about the interviewer was disclosed beyond their name and the name of their institution. Aside from the initial contact to explain this study’s objective, confirm the volunteer’s eligibility, and schedule the interview, there was little further communication before the interview. Pre-interview contact was limited to those who required technical assistance to join the video call. Only the interviewer and the interviewee were present during the interviews.

### 2.2. Data Collection

Data were collected via semi-structured interviews. To this end, the authors developed an interview guide on the basis of the reviewed literature (Figure A1). The guide was piloted with three nurses independent from this study [22]. The interviews were video-recorded via Microsoft Teams’ built-in tools. The researchers’ affiliated institution provided access to this application and a secure digital environment for recording the interviews. As an additional security measure, the recordings were transferred to an external drive within minutes of each interview’s conclusion. Notes were taken during and after the interviews in a field notebook. These notes included codes and reflexive annotations. The average interview duration was 60 min, with a maximum of 90 min and a minimum of 45 min. The interviews were conducted between February 2023 and April 2024. The final number of interviewed nurses was determined by the achievement of data saturation. Data saturation was considered to have been reached after no new codes emerged in the final five interviews conducted. The determination of data saturation was reached by consensus among the three researchers responsible for coding. The interviewees received a transcript of their interview to allow for comments or corrections. None of the participants expressed any wish to comment or correct anything.

### 2.3. Data Analysis

Three researchers (A.M.C-B, M.M-A and I.M-T) conducted a consensus-based qualitative data analysis following the logic of three-step inductive thematic analysis [28,33,34]. The researchers analysed the data independently and then discussed the findings until a consensus was reached [28,33]. Initially, units of meaning were identified and assigned one or more codes. These codes were then grouped by affinity into emerging categories. Finally, themes were identified from the emerging categories [28,29,33,35]. All three researchers were nursing graduates with experience in qualitative research and in the field of climate change and health. Reflexivity regarding the researchers’ influence on data analysis was constant throughout the process, both individually and collectively. The third researcher enabled triangulation in cases of discrepancy between the first two analysts [33]. In addition, the field notebook served as an additional source of data triangulation [28,35]. Data analysis was conducted via ATLAS.ti 25 software (ATLAS.ti Scientific Software Development GmbH, Berlin, Germany). The interviewed nurses were provided with preliminary results during the coding process. No comments were received regarding these preliminary results.

### 2.4. Ethical Considerations

The research protocol was approved by the Clinical Research Ethics Committee of the Cuenca Health Area (REG. 2021/PI3721). This study was designed in accordance with the Declaration of Helsinki. The volunteers were informed during the initial contact that the interview would be recorded and of their right to request the deletion of their interview material at any time during this study. Informed consent, both verbal and written, was obtained prior to the interview. The interview transcripts were anonymised by the same researcher who conducted them. Each interviewee received a copy of their anonymised transcript and of the preliminary results obtained during the coding process.

### 2.5. Quality and Rigour

Various strategies were followed to ensure the quality and rigour of this study. The research team engaged in a conscious effort of reflexivity throughout the entire research process. This reflexivity encompassed aspects such as the researchers’ influence on data collection and interpretation, as well as the influence of the chosen methodology on the responses obtained [31]. The reflexive process was shared among the researchers and cross-checked against notes recorded in the field notebook.

A purposive sample with heterogeneous characteristics was established to ensure representativeness [36]. The potential influence of contextual factors on interviewees’ responses was considered in the selection of areas with varying levels of pollution and different nursing specialities while ensuring homogeneity across other contextual aspects such as nationality, language, professional background, and employment in public institutions. The influence of both the contextual elements common to all participants and the variation across groups was explicitly addressed during the data analysis phase.

The adoption of a qualitative descriptive approach within a constructivist paradigm was a deliberate decision, aligned with the aim of exploring knowledge and perceptions as constructed through lived experience. The research team engaged in reflexive discussions regarding the potential influence of the chosen methodology on the findings. Methodological decisions were grounded in the relevant literature and subjected to critical review throughout the research process.

The potential influence of the interviewer on participants was also taken into account. There was no prior relationship between the interviewer and the interviewees. To minimise this potential influence, efforts were made to foster a relationship of openness and trust. A context of trust was created by allowing participants to choose the setting in which the interview took place, ensuring that no researchers other than the interviewer were present, and safeguarding the anonymity of the data. Furthermore, the participants were explicitly reassured of the value of all contributions, and active-listening techniques were employed throughout. Semi-structured interviews were conducted by the same researcher via a guide developed on the basis of the literature and previously piloted [22,28,33]. As a self-reflexive exercise, the interviewer recorded their reflections in a field notebook during and after the interviews. These notes were subsequently discussed among the researchers during the data analysis phase.

The researchers’ prior knowledge and experiences may also have influenced the data analysis process. Three expert researchers (A.M.C-B; M.M-A; I. M-T), all of whom are nurses and qualitative researchers, analysed the data independently and then discussed the findings until a consensus was reached [28,33]. This process of discussion among the researchers enabled the identification and critical examination of some of these potential influences. In addition, the interviewees received both a copy of the anonymised transcript of their interviews and the preliminary results to avoid bias [37].

The COREQ (Consolidated Criteria for Reporting Qualitative Research) guidelines were followed in the drafting of this manuscript [37].

## 3. Results

A total of 31 nurses from five different specialities within the public healthcare services of nine regions in Spain were interviewed. The sample consisted predominantly of women (n = 24, 77.42%), with a mean age of 41 years (range: 26–60). The average length of service experience was 12 years (range: 1–32), with 70.97% of participants having less than 15 years of experience (Table 1). The individualised demographic data can be found in Appendix B (Table A1).

Ten codes were identified and grouped into the four categories outlined below (Figure 1):

### 3.1. Category 1: General Knowledge of Climate Change

#### 3.1.1. Code 1.1: Causes of Climate Change

All the participants acknowledged the existence of climate change. Although ten interviewees considered the possibility of a natural origin, all agreed that—regardless of its root cause—climate change has been intensified and accelerated by human activity. The participants identified several key ways in which human activity has contributed to climate change. These included pollution, excessive resource consumption, the manner in which resources are utilised, and high levels of waste production.

When discussing pollution, they tended to use the term in a broad sense—referring to air, soil, or water—without making clear distinctions. It was common for them to conflate concepts such as pollution, waste, greenhouse gases, and ozone layer depletion.

“I think it’s a bit of everything. As human beings, we sort of assume the Earth belongs to us, and we do all sorts of outrageous things… with oil, with energy, with the sun, with electricity and all that. But also, I believe there’s a natural component from the planet itself.”(N21)

“Maybe it is a natural cycle, but mankind’s hand is speeding it up a little—with so many factories, all the planes, the transport, and all those things that, really, didn’t exist before.”(N2)

“I mean, the air pollution—which we ourselves are creating—yes, it does cause this climate change because we’re damaging the ozone layer and ending up with less natural protection from the planet.”(N21)

“Well, my perception is that it’s like a layer […] that we’re creating in the atmosphere through consumption, […] like cars, fuel, for generating energy—we’re not using renewable energy, I mean, it’s all gas burning […].”(N12)

#### 3.1.2. Code 1.2: Consequences of Climate Change

The increase in temperature was the most frequently cited consequence of climate change. Other effects mentioned included sudden shifts in extreme temperatures, droughts, and a general rise in pollution, though participants did not specify the type. They also noted increases in both the frequency and intensity of Saharan dust intrusions and extreme weather phenomena.

The collateral effects mentioned by the participants included disruptions in agricultural and fishing yields, food scarcity, and increased reliance on fertilisers and phytosanitary products to maintain production. The interviewees also highlighted glacial melting, rising sea levels, and an extended pollen season as consequences of high temperatures.

“That we reach much higher temperatures, that there’s no water in the rivers like we’re used to, […] and that, of course, affects the entire ecosystem, impacting other areas as well, you know? Microorganisms, animals, plants.”(N18)

“Well, pollution and, yeah, all these changes, many of which are already plain to see. The lack of rain, the extreme heat… These strange weather changes we’re having… And well, the melting of the polar ice caps.”(N16)

“It seems we’re experiencing periods of torrential rain, periods of drought… the global temperature has risen. […] It seems like we’re heading towards extreme climates. […] What we can say in Spain, […] is that the temperature is rising. […] And also, it’s very dry.”(N24)

#### 3.1.3. Code 1.3: Expectations Regarding Climate Change

According to the participants, the phenomenon of anthropogenic climate change is not a recent one, but rather has become increasingly evident in recent years. They expressed pessimistic views about the future, believing that the effects of climate change will continue to intensify and could ultimately render the Earth uninhabitable. Although they acknowledged current climate effects, they did not believe that they would personally witness the most severe consequences.

“We’ve been witnessing climate change for… for many years now. […] With phenomena that may be used to happen once every so often, and now we’re seeing them more and more. But it’s not something in the future. It’s something that’s happening now and that, with time, will get worse. It will get worse.”(N9)

“My first impression is that it’s unstoppable. That it’s been unstoppable for years now and that it’s been like the boy who cried wolf, hasn’t it? Always saying it’s coming, it’s coming—but people didn’t believe it, and now there’s no turning back.”(N19)

“I’m pessimistic about this in the long run. Will I selfishly not be around? Yes. And I hope so. Because otherwise, it would be unbearable.”(N9)

### 3.2. Category 2: Knowledge Regarding Climate Change and Health

#### 3.2.1. Code 2.1: Health Effects of Climate Change

The participants identified health deterioration as the primary consequence of climate change for humans. Respiratory issues were most frequently cited, with allergies and exacerbation of conditions such as asthma and COPD linked to increased pollen and pollution. This was followed by hyperthermia, dehydration, and heatstroke, which are attributed to rising temperatures. Mental health impacts, such as anxiety and depression, were also frequently mentioned.

Drought and pollution are perceived to negatively impact nutrition—both through reduced food access by contributing to food contamination. Extreme temperatures and weather events were associated with disruptions to daily life, including physical inactivity, isolation, sleep disturbance, economic hardship, spread of infections, and climate-induced migration. Less frequently mentioned effects included cancer—mainly linked to pollution—and dermatological issues—associated with ozone layer depletion.

“Lung diseases come first. […] Increases in cases that weren’t diagnosed before… Many types of cancer, […]. And anxiety… depressions… […] I imagine it’s due to air pollution, […] and then the temperature changes. […] Even climate change in the ‘calima’ [airborne dust] that comes from North Africa.”(N5)

“Temperature, drought, a change in how all these conditions affect people’s health. Heat strokes in older people, decompensations in people with heart failure. They decompensate far more in the summer months, far more. And mental health in general, especially among young people, right? This anxiety about the harsh conditions and not having any kind of future project.”(N18)

“The fact that it’s raining less, […] farming becomes more difficult, […]. To give you an example, […] the olive harvest was awful this year, so the price of olive oil will go up, […] and I’ll end up cooking with palm oil, which is much worse for your health, or with margarine, or another kind. I’ll be replacing the healthier oil—or at least the one I consider healthier—with one that’s cheaper.”(N22)

“It’s just really hard to sleep when it’s over thirty degrees. […] So, just imagine the sleep cycle—all those patients who already sleep poorly, all those who need benzodiazepines or anything else to sleep. If you add such high temperatures to that, it’s the perfect breeding ground for someone to throw themselves out the window. […] It should even be considered whether it really has an impact on, for example, bipolar patients, depressive patients, patients with any type of disorder.”(N8)

#### 3.2.2. Code 2.2: Vulnerable Populations

There was a clear perception that climate-related health impacts are already evident. Although all populations were considered potentially affected, the nurses identified particularly vulnerable groups: elderly individuals, chronically ill individuals, children, economically disadvantaged individuals, outdoor workers, and residents in specific areas.

“Because… any change affects people, but right now, in the case of an elderly person, a sudden rise in temperature brings about a whole range of changes—I mean, it affects all the systems in an older person’s body.”(N30)

“I don’t know… for example, older people or very young children—those who are more vulnerable—being exposed to high temperatures, well, that causes dehydration, doesn’t it? Just as an example.”(N27)

“It can also affect us, especially those who are more vulnerable, particularly elderly individuals, people who already have some chronic illness or who are immunosuppressed, cancer patients…”(N28)

### 3.3. Category 3: Knowledge Relating to Actions Against Climate Change

#### 3.3.1. Code 3.1: Responsibility for Action on Climate Change

The interviewees believed that action against climate change is possible, although they noted that such action is already overdue. Both mitigation and adaptation were viewed as essential. They consistently emphasised the importance of individual responsibility, stating that everyone can and should contribute from their own position. However, they stressed that a truly effective response must be collective. Governments, in particular, were seen as key actors through their leadership, policy development, and enforcement of global agreements. Nevertheless, confidence in governmental institutions taking on this responsibility was minimal. The participants expressed mistrust, pessimism, and frustration toward political leaders and international authorities.

“I mean, there’s no real intention behind anything […] it’s absolute hypocrisy, isn’t it? I mean, stop holding more climate change summits—you all meet up for four days, eat like pigs, everyone flies in by plane, and then you don’t reach any agreement, you know? Honestly, there’s nothing more hypocritical than climate change conventions—completely. […] And it’s not even because they’re not going to do anything—I’m not going to do anything either, you know? […] So you focus on what’s close to you and the group of humans you belong to—my family, my workplace, the population I care for… well, let’s see what we can do here, right?”(N18)

“The problem is that I can do my little bit, and everyone can do a little bit, but the issue is with the major powers or the people who actually have the ability to take action. I mean, these climate summits—how many years have they been going on? And what agreements have they reached? ‘No, we’ll postpone it to 2020.’ ‘No, let’s push it to 2035.’ Oh come on, for fuck’s sake! Postpone it to 2050 and by then we won’t even have a climate left, we won’t have anything. Sure, you as an individual can do small things, but if nothing is done on a large or macroeconomic scale, it doesn’t matter how much you do individually… does it?”(N25)

“On an individual level, I can do my bit, but that’s all it is—a grain of sand. So it needs to be large groups who are clear about this and truly convinced that we are influencing climate change. Until that happens, well, it’s difficult.”(N24)

#### 3.3.2. Code 3.2: Actions Against Climate Change

The participants were aware of several practical actions that could be taken to address climate change. Recycling was the most commonly cited action, followed by reducing the use of private vehicles; managing water, electricity, and heat usage; avoiding unnecessary purchases; using renewable energy; and opting for plant-based, local, and seasonal diets over meat products.

Most participants reported implementing several of these actions with varying degrees of consistency. However, their motivations often stemmed from economic savings, personal habits, upbringing, or health reasons rather than from environmental awareness. These motivations frequently resulted in nurses unconsciously transferring their personal habits into their workplace or consciously adapting them to their professional environment.

“I mean, waste management. Erm… well, electricity—I’m quite obsessive about lights being on when they’re not needed. Then water—responsible water use. […] Also… I don’t know, when it comes to consumption, I’m increasingly aware—or at least I try to be more aware—of responsible consumption. Right? Plastics… […]. Eating seasonal fruit, local fruit, […].”(N6)

“I recycle, I try to reduce energy use, try to cut down on petrol consumption, electricity, water. I try to do those little things—and to teach them to my daughter.”(N21)

“So, you try to cut down on consumption—also for one reason, because all these energy sources are so expensive that we’ve been forced to cut back, but because of our wallets. Not so much because we’re convinced of how harmful they are, but because it’s hitting our finances.”(N24)

“I don’t behave any differently at work than I do at home. Just like I have the habit of recycling at home, I recycle lids at work, I recycle the little glass jars, I recycle the plastic, I recycle the paper.”(N20)

### 3.4. Category 4: Development of Knowledge

#### 3.4.1. Code 4.1: Level of Knowledge

Although the level of knowledge about climate change varied considerably, most interviewees demonstrated a medium level of understanding. Nevertheless, they generally perceived their own knowledge to be low, regardless of its actual adequacy. Despite expressing interest in the topic, most did not actively seek out information. Their understanding came from conversations within their social circle, the media or social networks. Only a minority pursued further information from scientific or academic sources, typically only when prompted by something they had read or heard. Another key source of knowledge is the direct observation of climate change effects.

“What little I know comes from the news, what you see […]. My partner is often more interested in the topic and talks to me about it. […] Honestly, I haven’t picked up an article or a book to read about climate change—so it’s really just from news and the odd conversation that comes up.”(N28)

“I mostly do, well, maybe it’s more of an indirect search. I look up a lot of stuff about food and nutrition. […] So, as the phone and computer always show you related things, I end up getting news alerts and they catch my interest, and I click on them. No, it’s not like I actively search for it directly every day.”(N5)

“It’s something you come to notice through evidence—things you hear, things you read, things you see on the internet and in the news—and then you start to realise in your daily life, you think, oh right, it’s true.”(N31)

#### 3.4.2. Code 4.2: Motivations for Learning

The interviewed nurses expressed a clear interest in continuing to develop their knowledge about climate change, its effects, and the role of nursing. Their motivation primarily stemmed from personal interest in environmental issues, the preservation of natural surroundings, and the relationship between health and the environment. Other significant motivators included concerns for future generations and personal health. However, being a healthcare professional was not, in itself, a sufficient reason for further study on the topic.

“I would love to receive a session on climate change from the perspective we’re talking about here […] I mean, as a nurse and in the environment I work in, it wouldn’t have occurred to me—but yes, of course, yes.”(N7)

“But the thing is, there will still be people after me. I mean, for example, I’m not a mother, but other people will still be here.”(N9)

“And I think a lot about what we’re going to leave for our children, and I believe that… […] maybe not my children, but I’m not sure in what conditions my grandchildren will live.”(N5)

“Concern, information… mainly about things that affect health. If this affects me, why does it affect me, and what do I need to avoid so that it doesn’t?”(N20)

#### 3.4.3. Code 4.3: Barriers to Learning

Despite their interest, many nurses identified eco-anxiety as a significant barrier to engaging more deeply with the topic. Consideration of climate change has been shown to evoke a range of emotions, including feelings of distress, insecurity, pessimism, frustration, anger, fear, distrust, uncertainty, powerlessness, and disappointment. To avoid these unpleasant emotions, several participants admitted to consciously limiting their exposure to climate-related information.

“In general, my feeling is one of fear, uncertainty, and mistrust—not knowing what’s going to happen.”(N15)

“Honestly, it makes me really sad and really angry. Truly an overwhelming sense of helplessness.”(N8)

“I don’t go into it too deeply. So I don’t get overwhelmed either.”(N9)

“I try to read about the topic. Not too much, because to be honest, I end up feeling really hopeless seeing that nothing is improving. And sometimes I’d almost rather live in ignorance. That’s it. It’s harsh, but it’s the reality. It’s really because… because I feel so powerless seeing that we’re not moving forward… I’d almost rather not know anything.”(N5)

However, accessing reliable sources of knowledge is not straightforward. Notably, the current workplace, academic curriculum and professional association offerings do not include training on climate change. Consequently, nurses must independently seek out information, which exposes them to an overwhelming volume of content. This, if not effectively managed, can contribute to the development of eco-anxiety.

“I studied nursing from 2014 to 2018—climate change already existed, it was already a current issue—and I don’t remember any subject where it was discussed. So, they could have instilled the topic a bit more from university onwards. […] I think the problem is ignorance, a lot of the time.”(N28)

“INTERVIEWER: What’s the first thing that comes to mind when you hear ‘climate change’?

INTERVIEWEE: The first thing, mm, maybe anxiety. […] Anxiety, because it’s something that feels uncontrollable, right? And… and well, also because of the news—the way they report what’s happening with climate change. […] It’s like a bombardment of news, right? And told in a somewhat catastrophising way. […] I often try, well, not… or to stay informed, but maybe in another way. A radio show or headlines, but without going too deep.”(N4)

## 4. Discussion

This study aims to explore Spanish nurses’ knowledge of climate change and its effects on health. The interviewed nurses demonstrate a range of knowledge regarding climate change and its health implications, with the majority recognising its anthropogenic origins and associated health risks. The interviewed nurses demonstrate a higher level of knowledge than participants in other studies [21]. This may be explained by their residing in one of the regions of Europe most affected by climate change [5,38,39]. Overall, their perceptions of climate change are similar to those of health professionals in other southern European countries, such as Italy [24] and Portugal [40]. Despite their knowledge, and in line with other studies, our participants also conflate terms when referring to climate change [21,41] or overestimate natural causes as the origin of climate change [40,41]. Nonetheless, as also reported in larger-scale studies [42,43,44], the proportion of our sample that attributes climate change to natural causes is minimal. The interactions between different codes highlight the central role of knowledge level, upon which all other codes depend—except for those of motivators and barriers to learning, which either promote or hinder it, respectively (Figure 1).

### 4.1. Nursing Potential and Areas in Need of Strengthening

The interviewed nurses identify the groups most vulnerable to the effects of climate change. To varying degrees, nurses demonstrate the knowledge, capacity, and sense of climate justice necessary to map the groups most vulnerable to climate change within their environment [45]. These qualities make nurses essential figures in the design of mitigation and adaptation strategies. However, the heterogeneity of knowledge among nurses also becomes evident, highlighting the need to reinforce basic knowledge of climate change, including its anthropogenic origins.

### 4.2. The Influence of Demographic Data

No significant associations were observed between the demographic variables collected and the findings obtained. Notably, those nurses working in community settings demonstrate a heightened perception of economic vulnerability as a risk factor [41]. The provision of longitudinal care by primary care and geriatric nurses may allow for the identification of socioeconomic vulnerabilities, which are more challenging to discern than biological ones. Irrespective of the air quality zone, speciality, age, sex, or years of experience, the interviewees prioritise aspects that directly affect them—perceived empirically. They then consider those impacting their immediate environment, such as their workplace, city, or region—perceived through their social circle—and, finally, more distant consequences—known only through media and social networks.

### 4.3. The Significance of the Audience’s Context

Our findings reveal a strong contradiction between the local perception of certain aspects of climate change and the distant way in which others are conceived (e.g., the emphasis on local environmental problems throughout the interviews versus solutions to the climate emergency often expressed as dependent on distant entities such as national or supranational bodies). Among the nurses interviewed, this contrast appears to be linked to the sources of information they use. The influence of empirical perception on how intensely a phenomenon is perceived is well documented [46]. The influence of empirical perception and the local environment highlights the need to tailor training to each audience’s context to develop a planetary health perspective. This process should begin with a local frame of reference, leveraging personal interest to cultivate sustainable habits. Starting with local climate issues that are of significance to nurses facilitates the establishment of meaningful connections, thereby enabling the progression of their training towards a more comprehensive understanding of planetary health or a One Health approach. Berubé et al. [20] likewise underscore the potential influence of personal interests in their review of the scientific literature and official documents issued by nursing associations, which examine nurses’ engagement with climate change.

### 4.4. Informal Knowledge Sources and Their Consequences

The use of informal knowledge sources and the lack of formal training are common barriers [20,21,24,41]. According to a review conducted by Yeboah et al. [21], social media constitutes the primary source of information on climate change used by nurses. A survey of 1200 Japanese nurses reported that 73.8% of respondents had acquired their knowledge of climate change through mass media such as television [47]. The media serve as a powerful tool for raising awareness about the climate crisis but often provide biased information [48,49,50,51]. A bias that is often amplified by the media is the politicisation of climate change [48,51], which is widely regarded as a barrier to effective action [40,42,52]. Another common distortion is the media’s tendency to prioritise or exaggerate the most negative aspects of the issue [53,54,55], generating anxiety and rejection among the public [56]. The lack of formal training leaves nurses vulnerable to misinformation, hoaxes, or anxiety from information overload. This underscores the responsibility of educational institutions, workplaces, and professional associations to provide nurses with high-quality training on climate change.

### 4.5. Eco-Anxiety

The level of eco-anxiety expressed by the interviewed nurses is shared by more than half of the Spanish population, with the highest levels reported among women aged 60–69 with high educational attainment [57]. In our findings, those expressing the most eco-anxiety are women aged between 35 and 55. However, it should be noted that women in that age range make up the majority of our sample. Although eco-anxiety can produce symptoms similar to other forms of anxiety, in most cases, it cannot be considered pathological [58]. Rather than being a state to be avoided, eco-anxiety is frequently associated with greater engagement in pro-environmental actions [58,59]. It is not negative for nurses and society to feel anxious about the climate crisis. Nevertheless, it can be reframed as a potential driver for action. When eco-anxiety is correctly understood and supported, it has the potential to act as a catalyst, transforming emotional distress into motivation to respond to the climate emergency. For this to happen, it is vital to provide nurses with the right environment and tools to channel their emotions and transform them into a desired and healthy state of responsiveness to the climate crisis [58].

### 4.6. Leadership Gap

The need for greater governmental involvement is a view shared by both health professionals and the general population, at national and international levels [39,44,60,61]. This perception is also reflected in the articles included in the review conducted by Yeboah et al. [21]. Furthermore, a survey of 1022 Korean nurses observed the influence of leaders who take initiative. Nurses working in institutions where leadership had promoted climate action demonstrated greater awareness and knowledge of climate change [62]. The nurses express frustration with the lack of leadership and decisive measures from governments and political institutions. However, nurses themselves rarely take the initiative to fill this leadership gap, as the alternatives they propose tend to rely on individual action [20].

### 4.7. Individual Actions and Motivation

In addition to mere resignation in the face of a lack of collective action, nurses recognise the importance of individual responsibility. The interviewees mention individual actions against climate change similar to those described by other nurses [21]. While recycling is the most frequently mentioned action, only a few nurses mention the importance of reducing waste generation as a higher priority than recycling [63,64]. The main motivations for undertaking these individual actions are financial savings and habits acquired in the private sphere. Concern for future generations, also reported by Bérubé et al. [20], is another motivation mentioned by the interviewees. Savings as a personal motivator represent an opportunity to align individual motivations with institutional goals. The influence of habits acquired during childhood and in private environments indicates the need to expand nursing education beyond the professional context. Training nurses on climate change should focus not only on the health effects of climate change and the nursing role but also on instilling transversal habits applicable to personal life. Once integrated into individual routines, these habits may serve as catalysts for action within the workplace [27,65].

### 4.8. Limitations

One limitation of this study may be the number of participants. Nevertheless, the use of a qualitative methodology is a strength, as it enables deeper insight and prioritises the richness and saturation of data over sample size. Although the sampling techniques used might have encouraged the recruitment of participants with greater interest in the topic, this approach may have resulted in more enriching data. Despite the limited geographical and speciality representation, efforts were made to ensure sample representativeness by establishing a purposive sample with heterogeneous characteristics in terms of the air quality zone and speciality. Moreover, as a cross-sectional study, it is possible that external events may have influenced the responses. However, the interviews were conducted on different dates, alternating specialities and air quality zones over fourteen months, thereby minimising biases related to seasonal weather or media events. Furthermore, the choice of interviews as a data collection technique could have given rise to social desirability bias since anonymity between the interviewee and interviewer could not be maintained. Nevertheless, the interviews proved to be a highly effective tool for stimulating reflection among the nurses, as reported by participants in this and other studies [41]. To mitigate this limitation, efforts were made to establish a climate of trust. All interviews were conducted by the same researcher, who was also responsible for producing the transcripts. Interviews took place in locations chosen by each participant, without the presence of other researchers. The transcripts were anonymised before being shared with other members of the research team. Finally, the participants were invited to contact the research team should any concerns arise after reviewing the transcript of their interview or the preliminary results. No participants made contact. The absence of any participant responses was interpreted by the team, based on their experience, as an indication of consent for both the transcription and the preliminary results. Furthermore, the triangulation strategy was addressed through complementary means, including blind consensus coding by three researchers and reflexive engagement with the field notebook.

### 4.9. Implications for Clinical Practice

In clinical practice, nurses are not only able to address the health effects of climate change on the population but, as our results indicate, are also well suited to identifying vulnerable groups and mitigating the healthcare system’s carbon footprint. Our findings underscore the value of nurses’ holistic perspective in the planning and implementation of climate change mitigation and adaptation strategies. This highlights the opportunity to include nurses in public health policy development teams. However, nurses’ basic knowledge of climate change and its health effects must be strengthened. Nurses’ education should be developed within a planetary health and One Health framework. This training should be continuous, both pre- and postgraduate, and adapted to the specific competencies required to tackle the diverse impacts of climate change in different healthcare contexts. Academic institutions, healthcare centres, and professional organisations must actively promote this education.

Moreover, nurses should receive training in the skills needed to enhance their professional role, such as communication, research, negotiation, and leadership [66,67,68,69]. Nurses could address the leadership gap through greater political and social participation, for instance, by forming climate action groups such as Green Teams within healthcare institutions [27]. These groups would promote proactive integration of climate change into daily discussions. This will prevent the issue from being relegated to isolated training sessions or superficial media coverage [3,70].

In conclusion, our findings highlight both the barriers to and opportunities for the development of the educational curriculum for nurses, as well as for policy initiatives in response to climate change. We have identified pedagogical enhancements such as the integration of the planetary health perspective and the imperative to cultivate key competencies including communication and negotiation. Furthermore, we have underscored the importance of enhanced political engagement and the value of incorporating nurses into the planning and formulation of public health policies. It is our hope that these proposals will be duly considered by those responsible for educational programmes and other relevant decision-makers.

Future studies may direct their objectives towards exploring nurses’ self-perceived role in the face of climate change, the perceptions of barriers hindering nursing action in response to climate change, or the development of strategies to promote the leadership role of nursing in the context of the climate emergency. One concrete example could be a qualitative descriptive study exploring perceived barriers to nursing leadership in the context of climate change. A sample of particular interest for such a study might include nurses in managerial or leadership positions. Another potential avenue for future research could be a national survey examining Spanish nurses’ perceptions and knowledge regarding climate change. This quantitative study could refine its questionnaire based on the findings of our research, thereby addressing the limitations in generalisability inherent in our design.

## 5. Conclusions

Despite the variation in knowledge among participants, the nurses interviewed showed a medium level of understanding and a strong interest in further development. However, most of their knowledge came from informal sources. Therefore, it is crucial to reinforce and expand this understanding through formal training provided by academic institutions, healthcare centres, and professional associations.

Although further development of their competencies is needed, nurses’ understanding of the health–disease process and its social determinants, together with their psychosocial approach to adaptation, positions them as essential stakeholders in the formulation of public policies to address this challenge. By enhancing both the knowledge and skills necessary to channel eco-anxiety productively, it is possible to cultivate the full potential of nursing—including underdeveloped roles such as leadership in climate response. Strengthening nurses’ knowledge of climate change is therefore essential for building climate-resilient healthcare systems.

## Figures and Tables

**Figure 1 nursrep-15-00226-f001:**
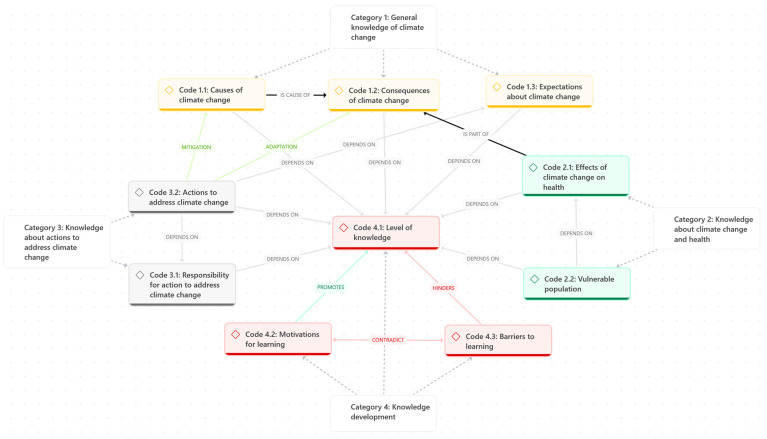
Code map.

**Table 1 nursrep-15-00226-t001:** Demographic data.

Variable	Category	Frequency	(Percentage)
Sex	Female	24	(77.42%)
	Male	7	(22.58%)
Age ^1^	20–29	4	(12.90%)
	30–39	10	(32.26%)
	40–49	9	(29.03%)
	50–59	5	(16.13%)
	60–69	1	(3.23%)
Speciality	Geriatrics	8	(25.81%)
	Primary care	6	(19.35%)
	Respiratory medicine	6	(19.35%)
	Cardiology	6	(19.35%)
	Emergency services	5	(16.13%)
Experience in the speciality	<5	11	(35.48%)
	5–15	11	(35.48%)
	>15	9	(29.03%)
Pollution level	High	9	(29.03%)
	Medium	11	(35.48%)
	Low	11	(35.48%)

^1^ Age was not recorded for two interviewees.

## Data Availability

The data are unavailable due to privacy and ethical restrictions.

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
