# Peer review of "Spanish Nurses’ Knowledge and Perceptions of Climate Change: A Qualitative Study"

_nursrep, 2025, doi:10.3390/nursrep15070226_

Round 1

Reviewer 1 Report

Comments and Suggestions for Authors

I thought overall this was an excellent paper of current great importance. 

I have a few minor niggles with the article, shown below:

Abstract: ‘nurses play a critical role in addressing climate change’….this is a sweeping statement without saying what their role is, and needs to show their role.

L40: it would be good if you said who you are quoting, rather than just the Vancouver citation number.

L72: could you enlarge on what ‘Green Teams’ are please? (OK, you do this in the discussion but some idea when you first mention it would be good).

Very clear aims.

Methods: any reasons for refusal to be interviewed after initial volunteering? And was the not fitting the inclusion criteria due to them not living in the areas selected? Wrong speciality? And why did you not select paediatric/neonatal services nurses as well as this group of patients is also more vulnerable.

I like how the quotations follow the commentary in the results.

Not sure about the quote L358-359; behind what?

Neat reporting in figure 1.

L469: do you mean ‘should assume’? Because if it is already happening, they don’t appear to be very successful.

Excellent discussion, but one thing that would be good to include is what do you as researchers bring to the study? I know the interviewer was a nurse, but how do your roles and knowledge impact the study?

References: please could they all be given in English for consistency.

Reviewer 2 Report

Comments and Suggestions for Authors
  1. Title & Running Head
  • The title suggests a single focus on “knowledge,” yet results and discussion also cover attitudes, emotions (eco‑anxiety), and perceived actions.
  • Consider adding “perceptions” or “perceived actions” to the title if you want to reflect the richer range of themes.
  1. Abstract
  • A few wording glitches: “underresearched” (missing space).
  • Results statement “barriers exist that affect their knowledge acquisition” is unclear.
  • Replace generic phrasing with two concrete example barriers (e.g., “lack of formal training, information overload”).
  1. Introduction
  • Several in‑text citation clusters list 3–4 references without synthesis, leaving the reader to infer key points.
  • The sentence “Southern European countries are experiencing the effects … more intensely” is not backed by specific figures or examples until later paragraphs.
  • Duplicate paragraph structure around “largest group of healthcare workforce”—trim.
  • Aim is stated at end, but research question could be crisper.
  • Tighten literature paragraphs: introduce a data point, interpret, then cite.
  • When citing multiple sources, indicate what they collectively show (e.g., “Studies X–Z consistently report knowledge gaps ranging from 40–70 %”).
  • Conclude intro with an explicit research question (e.g., “How do Spanish registered nurses perceive and acquire knowledge about climate change and its health impacts?”).
  1. Methods

4.1 Design, Paradigm & Registration

  • “Naturalistic and flexible approach” could be redundant—qualitative descriptive already implies this.
  • Add one sentence clarifying how the constructivist stance shaped interviewing and analysis (e.g., co‑construction of meaning, reflexivity).

4.2 Sampling & Participants

  • Rationale for selecting the five specialities could be explained (e.g., high exposure to climate‑sensitive conditions?).
  • 13 volunteers declined/failed inclusion—state reasons briefly to examine selection bias. Summarise main reasons for non‑participation.
  • No reflexive account of the interviewer’s positionality beyond name and degree (gender was given but not reflexively discussed).

4.3 Data Collection

  • Field‑notes use is mentioned but not integrated into analysis description. State whether field notes were coded or used for reflexive memos.
  • Microsoft Teams recording security/confidentiality briefly addressed?

4.4 Analysis

  • Thematic analysis citation (Guest, MacQueen & Namey) corresponds to applied thematic analysis, but you also cite Braun & Clarke later; clarify analytic framework to avoid mixing.
  • No discussion of intercoder reliability approach (beyond consensus).
  • Reflexive TA emphasises researcher subjectivity; include reflexivity statement.

4.5 Quality & Rigour

  • “Theoretical sample” language is more grounded‑theory; you used purposive—use consistent terminology.
  • Member‑checking: participants received transcripts but none responded; discuss limitations of non‑response.
  1. Results
  • Figure 1 code‑map is useful but low‑resolution in current draft; ensure high‑quality figure.
  • Category labels mix “knowledge” and “actions” (Categories 2 & 3); category 4 “Development of knowledge” includes both motivations and barriers—may need sub‑section headers for clarity.
  • Some quotes very long; shorten or use ellipses without losing meaning.
  • Pollution zones: table lists High/Medium/Low but there’s no definition of how zones determined.
  1. Discussion
  • Long paragraphs blend multiple arguments; consider subheadings (e.g., “Knowledge heterogeneity,” “Role perceptions,” “Eco‑anxiety as double‑edged”).
  • Discussion sometimes repeats results verbatim rather than interpreting.
  • Claims about “solid grasp of psychosocial approach” a bit overstated given findings; temper language.
  1. Limitations
  • Quote: “one limitation may be the number of participants” but then downplays; this is fine, but be explicit about how saturation was monitored (e.g., when last three interviews yielded no new codes).
  • Potential social‑desirability bias not discussed.
  • State saturation check method.
  1. Practical Implications & Future Research
  • Could include idea of integrating planetary health into CPD credits.
  • Future research suggestions broad; specify one qualitative (e.g., ethnography of green teams) and one quantitative (e.g., survey to test prevalence of eco‑anxiety).
  1. Conclusion
  • Phrase “solid grasp” again slightly over‑generalises.
  • Conclusion introduces new idea of “climate‑resilient healthcare systems” but not defined earlier.
  • Add one sentence defining climate‑resilient systems or cite WHO framework.
  1. References & Appendices
  • Spanish sources include URLs; check persistent identifiers.
  • Interview guide in English; clarify if interview conducted in Spanish and guide translated.
  • Add translator note for guide (“originally in Spanish; translated for appendix”).

Comments on the Quality of English Language
  • Multiple line‑number artefacts, spacing errors (“underresearched”).
  • Occasional long sentences; readability falls.
  • Use plain‑language edit pass; split long sentences.
  • Re‑check tense consistency (methods past tense, discussion present).

Reviewer 3 Report

Comments and Suggestions for Authors

Introduction: While the background is well-structured, consider broadening the international context and strengthening the theoretical framing, especially regarding constructivism’s influence on design and interpretation.

Methods: 

  • The manuscript mentions that three researchers conducted the analysis, but there is no mention of inter-coder agreement procedures. Add a sentence or two on how disagreements in coding were resolved and how consistency was ensured among coders.
  • Reflexivity is not addressed in the context of data interpretation. Include a brief reflection on how researchers' backgrounds, beliefs, or prior knowledge may have influenced theme development or interpretation.
  • While authors mention that data saturation was reached, the criteria or indicators used to determine this are not described. Explain how the authors assessed saturation (e.g., no new themes emerging, redundancy in responses).

Results: Well-presented. The thematic organization is strong. Consider integrating brief reflections on how participant demographics (Appendix A2) may relate to thematic trends.

Discussion and Conclusions: Authors might consider integrating a few more international nursing studies on climate knowledge for broader context, particularly outside of Europe. This could enhance the generalizability and comparative perspective of your discussion.

Consider elaborating on how findings may inform curriculum development or policy advocacy.

Figures and Tables: Figure 1 (code map) is effective. Slight improvements to labeling or visual clarity could enhance accessibility.

Comments on the Quality of English Language

The manuscript is readable and coherent but would benefit from final professional language polishing to improve flow, tone, and clarity.

Round 2

Reviewer 2 Report

Comments and Suggestions for Authors

Authors have addressed all the comments.